# Low-Dose Immunotherapy: Is It Just an Illusion?

**DOI:** 10.3390/biomedicines11041032

**Published:** 2023-03-27

**Authors:** Fausto Meriggi, Alberto Zaniboni, Anna Zaltieri

**Affiliations:** 1Oncology Department, Poliambulanza Foundation Hospital Institute, Via Leonida Bissolati 57, 25124 Brescia, Italy; 2Pharmacy Unit, Poliambulanza Foundation Hospital Institute, Via Leonida Bissolati 57, 25124 Brescia, Italy

**Keywords:** immunotherapy, financial toxicity, low-dose

## Abstract

The development and use of immunotherapy in the last decade have led to a drastic improvement in results in the onco-haematological field. This has implied, on the one hand, the need for clinicians to manage a new type of adverse event and, on the other hand, a significant increase in costs. However, emerging scientific evidence suggests that, as with other drugs in the recent past, the registry dosage can be drastically reduced for immunotherapies without penalizing their effectiveness. This would also lead to an important reduction in costs, expanding the audience of cancer patients who could access immunotherapy-based treatments. In this “Commentary”, we analyze the available evidence of pharmacokinetics and pharmacodynamics and the most recent literature in favor of low-dose immunotherapy.

## 1. Introduction

Although the use of immune checkpoint inhibitors (ICIs) has radically changed the treatment and outcomes of numerous types of cancer, it has also resulted in several new immune-related adverse events (irAEs) to manage, and a further exponential increase in direct and indirect costs. Researchers are exploring new strategies to address what can be called a real political–social–economic emergency, now well identified and called, without euphemisms, “financial toxicity” of cancer treatments. The price, not only in economic terms, that one risks having to pay is very high, in the sense that a large part of the world’s population, even in the highest-income countries, could remain excluded from access to the most innovative cancer treatments. A well-established, albeit far from decisive, alternative undertaken in an attempt to reduce costs is the marketing of biosimilar and generic drugs. Another viable option of great importance would be to reconsider, for the same effectiveness, the authorized dosage of some medicines, especially those with higher costs, especially when used on a large scale, such as those indicated for the treatment of most cancers. A practical example today is immunotherapy (IO).

## 2. Pharmacokinetic and Pharmacodynamic Rationale

Nivolumab (N) is a human immunoglobulin G4 (IgG4) monoclonal antibody (HuMAb) that binds to PD-1 (Programmed cell death protein 1) and blocks the interaction of PD-1 with both PD-L1 (Programmed death-ligand 1) and PD-L2 by enhancing T cell responses, including anti-tumor responses. In vitro tests showed the ability of N to greatly stimulate T cell responses and cytokine production in mixed lymphocyte reaction and superantigen, or cytomegalovirus stimulation, tests. With the use of N and target-activated T cells, no in vitro antibody-dependent cytotoxicity of cell-mediated or complement immunity was observed [1]. N was first approved by the Food and Drug Administration (FDA) in December 2014 for the treatment of metastatic melanoma at the dosage of 3 mg/kg administered every 2 weeks. Subsequently, in 2016, the recommended dosage was changed to 240 mg every 2 weeks and, in 2018, to 240 mg every 2 weeks or 480 mg every 4 weeks (flat dose) [2].

However, there has been significant pharmacokinetic (PK) and pharmacodynamic (PD) evidence for years to support the argument that N may work equally well, and probably with a better immune-related toxicity profile, even at significantly lower doses than those approved [3,4,5,6,7,8,9,10] In a phase I study conducted in 2012, N = 0.1 mg/kg (about 3% of 3 mg/kg) every other week showed activity and ability to saturate receptors similarly to higher dosages [11,12]. This would testify that there is most likely no dose–response correlation and that N could be administered at doses considerably lower than those currently used without affecting the results [13,14].

A recent analysis evaluated the PK exposure and clinical safety of N at flat-dose = 480 mg every 4 weeks compared to flat-dose = 240 mg every 2 weeks and compared to dose = 3 mg/kg every 2 weeks using quantitative clinical pharmacology approaches and safety data extrapolated from four phase III clinical trials. We found comparability of mean steady-state (SS) concentration over time between N = 480 mg every 4 weeks and 3 mg/kg every 2 weeks or 240 mg every 2 weeks, with similar pharmacokinetic exposures between different tumor types. The aggregated safety data from the CheckMate 066, 025, 057 and 017 studies for 61 patients (pts) who changed their dosage from N = 3 mg/kg every 2 weeks to 480 mg every 4 weeks were consistent with the safety profile of N = 3 mg/kg every 2 weeks and 240 mg every 2 weeks [15]. In a retrospective study conducted in Korea, pts data were analyzed with advanced non-small cell lung cancer (aNSCLC) treated with N in two centers. Pts who could not afford standard treatment for economic reasons received low-dose N (fixed dose of 20 mg or 100 mg every 3 weeks). The others received a standard dose of 3 mg/kg every 2 weeks. It was hypothesized that low flat-dose = 100 mg or = 20 mg could be effective based on phase I studies conducted on N that showed a response rate (RR) of 29% and 33.3% with N = 0.1 mg/kg and N = 1 mg/kg, respectively. Among the 47 pts with aNSCLC, 18 had received low-dose N. PD-L1 positivity was present in 13 (27.7%) pts and did not differ between groups. The RR was 13.8% in the standard dose group and 16.7% in the low-dose group (*p* = 0.788). N-dose or PD-L1 expression did not significantly affect progression-free survival (PFS) and overall survival (OS). Although the present study has several limitations (this is a retrospective analysis, the selection of the low-dose group was based on the economic status of the pts, and the small size of the sample has a low statistical power), it suggests the effectiveness of N even at a low dosage and could today represent a valid alternative option to reduce “financial toxicity”. Further well-designed prospective studies, such as a phase III non-inferiority study with a sample of sufficient size, are needed to confirm the efficacy of low-dose IO [6].

The same concept was observed for pembrolizumab (P), another anti-PD-1 monoclonal antibody widely used today in cancer treatments [16,17,18]. In an observational retrospective study with 114 pts with aNSCLC, Low et al. evaluated P at the fixed dose = 100 mg versus the standard dose = 200 mg, alone or in combination with chemotherapy, and found no significant differences in PFS and OS, nor in toxicity between the two doses [16]. Another study with P performed in Taiwan in aNSCLC pts showed that OS and irAEs percentages were comparable for both the standard dose (≥2 mg/kg) and low-dose (<2 mg/kg) P group. In addition, it showed that those pts who had received a dosage of P ≥ 1.8 mg/kg had obtained a better OS than those who had received a dosage of P < 1.8 mg/kg. These results suggest that a dose of P ≥ 1.8 mg/kg could represent the “clinically minimally efficient dose” [18], lower than the record flat-dose = 200 mg every 3 weeks or 400 mg every 6 weeks.

Based on the model and simulation of dose/exposure ratios for the efficacy and safety of P, there would be no clinically significant difference between a flat-dose = 200 mg every 3 weeks, a dose = 2 mg/kg every 3 weeks, or a flat-dose = 400 mg every 6 weeks [16,19]. According to the results presented by Freshwater et al., both the dosage = 2 mg/kg every 3 weeks and the flat-dose = 200 mg every 3 weeks would be appropriate for P, with neither regimen providing a significant PK advantage over the other. All pts, including those of higher weight, would be able to achieve P exposures within the range that has been shown to provide near-maximal efficacy in clinical trials. Exposures that correspond to or exceed those at 2 mg/kg ensure maximum target saturation is achieved, as indicated by the initial PK/PD KEYNOTE-001 work with a clinical biomarker (release of Interleukin-2), and where maximum target serum engagement at a dose of P = 1 mg/kg every 3 weeks has been demonstrated. It was also noted that the estimated median clearance of P in pts receiving 200 mg every 3 weeks was = 0.22 L/day, almost analogous to that obtained in pts receiving the dose of 2 mg/kg every 3 weeks (=0.23 L/day), as well as the clearance of endogenous IgGs (=0.21 L/day). All linear clearances (CL) were consistent with the typical CL characteristics of monoclonal antibodies (mAbs) (0.2–0.5 L/day) [20]. Moreover, SS simulations of a post-registration study involving P revealed a 90% probability of reaching the target of at least 99.31% for a 70 kg pt treated with a 200 mg regimen every 3 weeks, while a 1 mg/kg regimen every 3 weeks had a 90% probability of reaching 96.8% of the target. Therefore, a regimen of 1 mg/kg every 3 weeks (minimum C, SS, 12.8 µg/mL) versus a regimen of 2 mg/kg every 3 weeks (minimum C, SS, 25.5 μg/mL) can reasonably be expected to result in only a modest reduction in activity. In addition, it was noted that no difference in the exposure–response relationship was observed between doses of 1 to 10 mg/kg and therefore it was suggested that a regimen of 1 mg/kg every 3 weeks might be sufficient to achieve clinical efficacy [21].

Recently and once again, IO has shown efficacy and safety, even when administered at much lower doses than originally authorized [22]. Specifically, this was a phase III randomized trial that explored low-dose N treatment (20 mg flat-dose every 3 weeks) associated with methotrexate-based metronomic chemotherapy (MTX) plus celecoxib and erlotinib (TMC-I) versus TMC alone in 151 pts with a good ECOG (Eastern Cooperative Oncology Group) performance status (0–1) and carriers of relapsed or newly diagnosed head and neck (H&N) squamous cell carcinoma (SCC). The addition of low-dose N led to an improvement in 1-year OS from 16.3% to 43.4% (*p* = 0.0036). The median OS in the TMC and TMC-I arms was 6.7 months and 10.1 months, respectively (*p* = 0.0052) and the percentage of grade ≥ 3 adverse events was 50% and 46.1% in the TMC and TMC-I arms, respectively (*p* = 0.744). However, it should be noted that a share of enrolled pts in the study was treatment-naive, pts for which the addition of low-dose N may not be necessary, resulting in an overestimation of the outcomes. In addition, the control arm with low-dose MTX and erlotinib cannot be considered standard, especially in Western countries [23]. However, these conclusions have a socio-economic relevance that goes far beyond their scientific implications.

## 3. Discussion

The work of Patil et al. showed for the first time in a randomized phase III trial how a very low-dose IO (N = 20 mg flat-dose every 3 weeks) in addition to a regimen considered standard was both endowed with efficacy and well tolerated. The authors, administering only 6% of the usually recommended dose of N in the treatment of pts with advanced H&N SCC in addition to the TMC regimen, achieved a 1-year OS increase of 25%. These results, beyond the strengths and limitations of the study, require deep reflection, especially regarding the process that leads to the choice of the dosage of an active drug before being placed on the market.

A retrospective study was conducted in Canada to assess the economic impact of drug waste and explore more cost-effective dosing regimens and P preparation modalities. According to this work, having the possibility to use lower-dose vials, practice sharing in the preparation of P between various Antiblastic Drugs Units (UFA), and use weight-based dosing could have an overall positive impact on costs, although more research is needed, aimed at optimizing drug consumption and reducing waste [24].

Based on the additional evidence of PKs and pharmaco-economics [25], we, therefore, wanted to hypothesize the saving of P and N vials if “low-dose” regimens were used and not those with the “flat-dose” [Table 1].

Taking, for example, the 2022 calendar year, and assuming treatment of two adult pts with an average body weight of 75 kg with the flat-dose of P = 200 mg every 3 weeks, approximately 16 administrations would be planned and then 64 vials of 100 mg of P would be used, for a total annual administration for two pts = 6400 mg. If, on the other hand, we assume that we treat the same two pts with the dose of P = 2 mg/kg, the dosage of P would be 150 mg every 3 weeks. Approximately 48 vials of 100 mg of P would be used in one year which, compared to the 64 vials used for the “flat-dose”, would result in a “saving” of 16 vials per year of P.

If, on the other hand, a dosage of P = 1 mg/kg was prescribed every 3 weeks for four pts with an average body weight of 75 kg, the savings would amount to about 80 vials of 100 mg per year [Table 1].

The same considerations could apply to N. In a study conducted in Korea, Yoo SH et al. [5] showed that low-dose N can be equally effective in aNSCLC and could represent a viable alternative to contain “financial toxicity”. This logic could then also be transferred to other cancers usually treated with ICI. Thus, if the considered dose of N was not the flat-dose = 240 mg every 2 weeks, but = 20 mg or 100 mg every 3 weeks, regardless of the weight of the pt, the annual savings would be considerable. On the market, the currently available formats are doses of 40 mg, 100 mg, and 240 mg. Administering a dose of N = 3 mg/kg every 2 weeks would result in significant savings if the pt’s weight was around 70 kg.

For example, in our Oncology Department, considering the 2022 calendar year, for 4 pts weighing 70 kg, approximately 96 vials of 240 mg (=24 administrations for a total of 23,040 mg) would be used if the prescribed dose was the flat-dose = 240 mg every 2 weeks. If we assume, instead, to treat the same 4 pts with N at the dose = 3 mg/kg and with the same weight of 70 kg, the dosage every 2 weeks would be = 840 mg (=20,160 mg total per year). One could therefore argue for using vials with a content of 100 mg and 40 mg and not 240 mg, while also trying to optimize the administration by combining several pts with different dosages on the same day [Table 1].

Finally, the savings for both drugs would be more substantial if there was the possibility of organizing “drug day” programs in the Oncology Departments, which is the management and administration of a certain drug only on a specific day of the week to optimize the use of the packages by limiting waste. In conclusion, perhaps the simple question we should ask ourselves is, “Does the dose or regimen with which that active drug is marketed represent what is best for the pt in terms of efficacy and safety?” The “old” principle of Maximum Tolerated Dose (MTD) in phase I studies must now be superseded, and perhaps we should work on a new emerging concept, namely that of “minimum effective dose”, much closer to clinical and real-world practice. The scientific literature available to date shows that too often there are post-dose modifications of the approved regimens for several anticancer drugs, such as anti-PD-1 or anti-PD-L1 antibodies, without negative repercussions on efficacy and costs, while there is often a gain in terms of safety and economic savings. We also know of similar evidence transferable to numerous other anticancer drugs with the possibility of reducing the approved dose, such as for abiraterone acetate in prostate cancer [26,27,28]. In clinical practice, there is today a strong scientific and social rationale to support the further development of low-dose ICI-containing regimens, and we fully share the principles that motivate these studies, especially in an era of increasing economic hardship. For humanity and for the same effectiveness, is it better today to choose to treat many with “less” or to continue in wanting to treat a lucky few with “more”? Certainly, the answer is very complex, but we feel we are saying utopianly that, at least, people’s health should not be governed by purely commercial logic.

## 4. Conclusions

The increasingly strong and plentiful scientific evidence of PK/PD, together with the “real world” clinical practice, leads us to believe more than ever that the low-dose IO can become the standard to be administered in numerous types of cancers. Nevertheless, many clinical observations supporting/suggesting wider applications of low-dose immunotherapy in cancer treatments are based on small sample sizes of patients and, clearly, further studies are needed to define the best dosage for equal efficacy and toxicity. For all the social and economic benefits that would follow, we hope that it can soon become established clinical practice not only in developing countries but throughout the world, and that this too will not remain just an illusion.

## Figures and Tables

**Table 1 biomedicines-11-01032-t001:** Examples of savings with low-dose immunotherapy versus flat-dose immunotherapy.

N. of cycles/year with P 200 mg q3W (2 pts)	N. of vials/year of P 200 mg q3W (2 pts of 75 kg)	Dosage if P 2 mg/kg q3W (2 pts of 75 kg)	N. of vials/year if P 2 mg/kg q3W (2 pts of 75 kg)	Savings %
32	64	150 mg × 2 pts = 300 mg	48	25%
N. of cycles/year with P 200 mg q3W (4 pts)	N. of vials/year of P 200 mg q3W (4 pts of 75 kg)	Dosage if P 1 mg/kg q3W (4 pts of 75 kg)	N. of vials/year if P 1 mg/kg q3W, (4 pts of 75 kg)	Savings %
64	128	75 mg× 4 pts = 300 mg	48	>50%
N. of cycles/year with N 240 mg q2W (4 pts)	N. of vials/year of N 100 mg and 40 mg if N 240 mg q2W (4 pts of 70 kg)	Dosage if N 3 mg/kg q2W (4 pts of 70 kg)	N. of vials/year of N 100 mg and 40 mg if N 3 mg/kg q2W (4 pts of 70 kg)	Savings % (Vials of 40 mg)
96	192 (100 mg) + 48 (40 mg)	210 mg × 4 pts = 840 mg	192 (100 mg) + 24 (40 mg)	50%

P = Pembrolizumab; N = Nivolumab; pts = patients.

## Data Availability

Not applicable.

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
