# Peer review of "Low-Dose Immunotherapy: Is It Just an Illusion?"

_biomedicines, 2023, doi:10.3390/biomedicines11041032_

Round 1

Reviewer 1 Report

The Authors carefully analyze the available evidence of pharmacokinetics and pharmacodynamics in favor of low-dose immunotherapy. Their considerations are based on the current literature on the topic. This is an important topic for society because cancers are still a leading cause of morbidity and mortality worldwide. The commentary is well-thought out and well-organized. The conclusions are correct. Generally, I have not found significant limitations in this work; conversely, I think that it has many strengths, such as originality and accurate presentation of the issue.

Some minor remarks:

- Lines 23 – 24: This sentence is a bit unclear. I suggest rewriting it.

- Line 36: Explain the meaning of ‘PD-1’ at the first use. The same for ‘ECOG’ in line 121. 

- Line 41: Check the reference number (11 or 1?).

- Line 123: Lack of spaces before ‘The addition …’.

- You included a lot of abbreviations and I suggest making a list of them for better clarity.

Author Response

Response to the reviewer’s  1. Comments:

Every corrections suggested (lines 23-24,36,121,41,123) has been fixed and reported in red

Abbreviations has been now extensively explained  at the first use as suggested.

Reviewer 2 Report

The authors' commentary highlights a problem of some importance on low doses in oncological immunotherapy. And this problem could be extended to other branches of clinical practice.

Some queries need to be addressed:

a) (rows 43-45) The authors should specify that the recommended dosage of "240 mg every two weeks" and "480 mg every four weeks" are flat dosages, and they should add a reference.

b) Are there any official reasons why the FDA changed the administration plan beyond those "suspected" by the authors?

c) The studies cited by the authors are based on a small-size sample. Do the authors not think the studies could yield erroneous conclusions?

d)  The authors should also report the opposite opinion if any.

Author Response

Response to the reviewer’s 2 comments:

a) Flat dosages are now clarified and properly referenced (ref 28,added)

b) The “official reason” was that a  flat dose of 240 mg was selected as a harmonized dose across all countries  to facilitate global development of nivolumab monotherapy across tumor types and to facilitate and shortening of  the drug  preparation time by the Pharmacists.

c) A statement outlining the limitations of our paper,mainly based on the small-size sample of pts,has been added in the conclusions. (in red)

d) Opposite opinionsabout this issue are still lacking in the scientific literature (not surprisingly for us….)

Round 2

Reviewer 2 Report

The authors addressed the queries, even if not exhaustively.